

# Comprehensive profiling of retroviral integration sites using target enrichment methods from historical koala samples without an assembled reference genome

Pin Cui[1,*], Ulrike Löber[1,2,*], David E. Alquezar-Planas[1], Yasuko Ishida[3], Alexandre Courtiol[4], Peter Timms[5], Rebecca N. Johnson[6], Dorina Lenz[7], Kristofer M. Helgen[8,9], Alfred L. Roca[3], Stefanie Hartman[2] and Alex D. Greenwood[1]

[1] Department of Wildlife Diseases, Leibniz Institute for Zoo and Wildlife Research, Berlin, Germany
[2] Institute of Biochemistry & Biology, University of Potsdam, Potsdam, Germany
[3] Department of Animal Sciences, University of Illinois at Urbana-Champaign, Urbana, IL, United States
[4] Department of Evolutionary Ecology, Leibniz Institute for Zoo and Wildlife Research, Berlin, Germany
[5] University of the Sunshine Coast, Sippy Downs Queensland, Australia
[6] Australian Centre for Wildlife Genomics, Australian Museum, Sydney, Australia
[7] Department of Evolutionary Genetics, Leibniz Institute for Zoo and Wildlife Research, Berlin, Germany
[8] National Museum of Natural History, Smithsonian Institution, Washington, DC, USA
[9] Department of Veterinary Medicine, Freie Universität Berlin, Berlin, Germany
[*] These authors contributed equally to this work.

Corresponding author
Alex D. Greenwood,
greenwood@izw-berlin.de

## ABSTRACT

**Background.** Retroviral integration into the host germline results in permanent viral colonization of vertebrate genomes. The koala retrovirus (KoRV) is currently invading the germline of the koala (Phascolarctos cinereus) and provides a unique opportunity for studying retroviral endogenization. Previous analysis of KoRV integration patterns in modern koalas demonstrate that they share integration sites primarily if they are related, indicating that the process is currently driven by vertical transmission rather than infection. However, due to methodological challenges, KoRV integrations have not been comprehensively characterized.

**Results.** To overcome these challenges, we applied and compared three target enrichment techniques coupled with next generation sequencing (NGS) and a newly customized sequence-clustering based computational pipeline to determine the integration sites for 10 museum Queensland and New South Wales (NSW) koala samples collected between the 1870s and late 1980s. A secondary aim of this study sought to identify common integration sites across modern and historical specimens by comparing our dataset to previously published studies. Several million sequences were processed, and the KoRV integration sites in each koala were characterized.

**Conclusions.** Although the three enrichment methods each exhibited bias in integration site retrieval, a combination of two methods, Primer Extension Capture and hybridization capture is recommended for future studies on historical samples. Moreover, identification of integration sites shows that the proportion of integration sites shared between any two koalas is quite small.

Subjects Bioinformatics, Evolutionary Studies, Genomics, Virology
Keywords Integration sites, Retroviral endogenization, KoRV, Target enrichment, Clustering

## INTRODUCTION

Vertebrate endogenous retroviruses (ERVs) descend from exogenous retroviruses that infected the ancestral germ line and have subsequently been transmitted vertically from parent to offspring through Mendelian inheritance (*Coffin, Hughes & Varmus, 1997*). ERVs comprise up to 8–11% of vertebrate genomes (*Bromham, 2002*; *Pontius et al., 2007*). Most ERVs colonized their host genomes millions of years ago (*Khodosevich, Lebedev & Sverdlov, 2002*; *Gifford & Tristem, 2003*) making it difficult to study the process of retroviral invasion. Among vertebrates the exceptions are the EAV-HP virus of chicken and the koala retrovirus (KoRV) which spreads both horizontally and vertically among koalas *(Phascolarctos cinereus)* (*Sacco & Venugopal, 2001*; *Tarlinton et al., 2005*; *Tarlinton, Meers & Young, 2006*; *Simmons et al., 2012*; *Wragg et al., 2015*), and unlike most other described ERVs, are still in the process of endogenizing into the germ line of the host species (*Tarlinton, Meers & Young, 2008*). Therefore, in mammals, KoRV provides a unique opportunity to study the processes underlying ongoing retroviral endogenization. Historical DNA analysis from museum koala samples collected during the 19th and 20th centuries demonstrated that KoRV was already ubiquitous in northern Australia by the 19th century (*Ávila-Arcos et al., 2013*), and that the KoRV genome has remained strongly conserved (*Tsangaras et al., 2014b*). In contrast, KoRV integration sites among individuals are highly variable (*Tsangaras et al., 2014b*; *Ishida et al., 2015*).

Identical regulatory sequences at the 5′ and 3′ ends of the proviral genome, designated long terminal repeats (LTRs), are used to mediate viral integration within a host. The distribution of retroviral integration sites in the host genome is generally regarded as non-random (*Cereseto & Giacca, 2004*), with several factors influencing integration site selection, including viral integrase (*Lewinski et al., 2006*) and host chromosomal features (*Santoni, Hartley & Luban, 2010*). Retroviruses belonging to the same group tend to exhibit similar integration site preference (*Mitchell et al., 2004*; *Kvaratskhelia et al., 2014*). For example, gammaretroviruses in particular have been shown to preferentially integrate into the vicinity of enhancers, gene promoters and CpG Islands (*LaFave et al., 2014*). Despite these tendencies in integration site preference, the integration of a retrovirus within a precise location in the host genome is still a random event. All individuals in a host population may share older ERV integration sites as they become fixed in the population over time through drift, as is now true for most human endogenous retroviruses (*Blikstad et al., 2008*). In contrast, if a retrovirus endogenized very recently, the integration site will be rare among all but related individuals such as offspring, as is the case for KoRV among koalas (*Tsangaras et al., 2014b*; *Ishida et al., 2015*). The comprehensive identification of ERV integrations within host genomes would allow for research on how ERVs are affected over time by drift, selection and gene flow. Although KoRV integration sites have been examined in koalas, previous studies have not attempted a comprehensive survey of integration sites within or between host individuals. The focus of the current study was to evaluate methods that may comprehensively characterize retroviral integrations and which could be applied to museum samples to examine historical trends in the frequency of shared and unique KoRV integration sites.

Inverse PCR has conventionally been used for retrieving retroviral integration sites (*Nowrouzi et al., 2006*). Methods such as rapid amplification of cDNA ends (RACE), ligation-mediated PCR, linker-selection-mediated PCR, linear amplification–mediated PCR and genome walking (*Bushman et al., 2005*; *Moalic et al., 2006*; *Schmidt et al., 2007*; *Kustikova, Modlich & Fehse, 2009*; *Hüser et al., 2010*; *Ciuffi & Barr, 2011*) have also been used. However, it is unclear whether these methods can comprehensively detect integration sites given the potential for primer-target mismatch, and they have never been applied to ancient DNA (aDNA). DNA extracted from museum samples has the characteristics of aDNA, e.g., it is heavily fragmented (with most molecules shorter than 300 bp), damaged (e.g., uracil deamination), and in extremely low concentration (*Willerslev & Cooper, 2005*). The DNA degradation, fragmentation and contamination that occurs post mortem makes aDNA research technically challenging (*Pääbo et al., 2004*; *Allentoft et al., 2012*), often preventing the use of conventional molecular biological methods such as PCR.

To overcome the limitations of working with historical DNA, we applied three target enrichment techniques followed by high-throughput Illumina sequencing. The three techniques, Single Primer Extension (SPEX) (*Brotherton et al., 2007*), Primer Extension Capture (PEC) (*Briggs et al., 2009*) and hybridization capture (*Maricic, Whitten & Pääbo, 2010*) have been applied successfully to aDNA and could potentially be employed to determine sequences flanking targeted ERVs. Although inherently different, both SPEX and PEC are amplification techniques that specifically target a template strand at a locus of interest. The primer in each case will extend until physically halted or until the end of the template molecule is reached. By contrast, hybridization capture represents a range of varying methodologies used to enrich target sequences by 'capturing' the desired target sequence using hybridization to pre-designed probes. In all three methods, unwanted non-target molecules are washed away, while the enriched template is subsequently re-amplified before high-throughput sequencing. For a detailed overview of the three methods see Fig. 1. Ten koala museum samples collected between the 1870s and the 1980s were successfully examined. Because no assembled koala genome is currently available, an assembled host-reference-independent computational pipeline was established. The primary aim of this study was to compare the enrichment capabilities of these three methods with respect to establishing the number of KoRV integration sites retrieved from ten museum koalas. We additionally sought to determine the number of integration sites that were shared across koalas or unique to one koala, and to compare our results to those of published studies on integration sites in historical and modern koalas.

## MATERIALS & METHODS

### Samples and ancient DNA extraction

A total of thirteen museum samples were examined (Table 1). DNA extractions were performed in the ancient DNA (aDNA) laboratory of the Department of Wildlife Diseases of the Leibniz Institute for Zoo and Wildlife Research in Berlin, Germany. The laboratory is dedicated to aDNA work and has never been used for molecular work on modern samples. The room is UV irradiated 4 h every night by ceiling-mounted UV lights. All

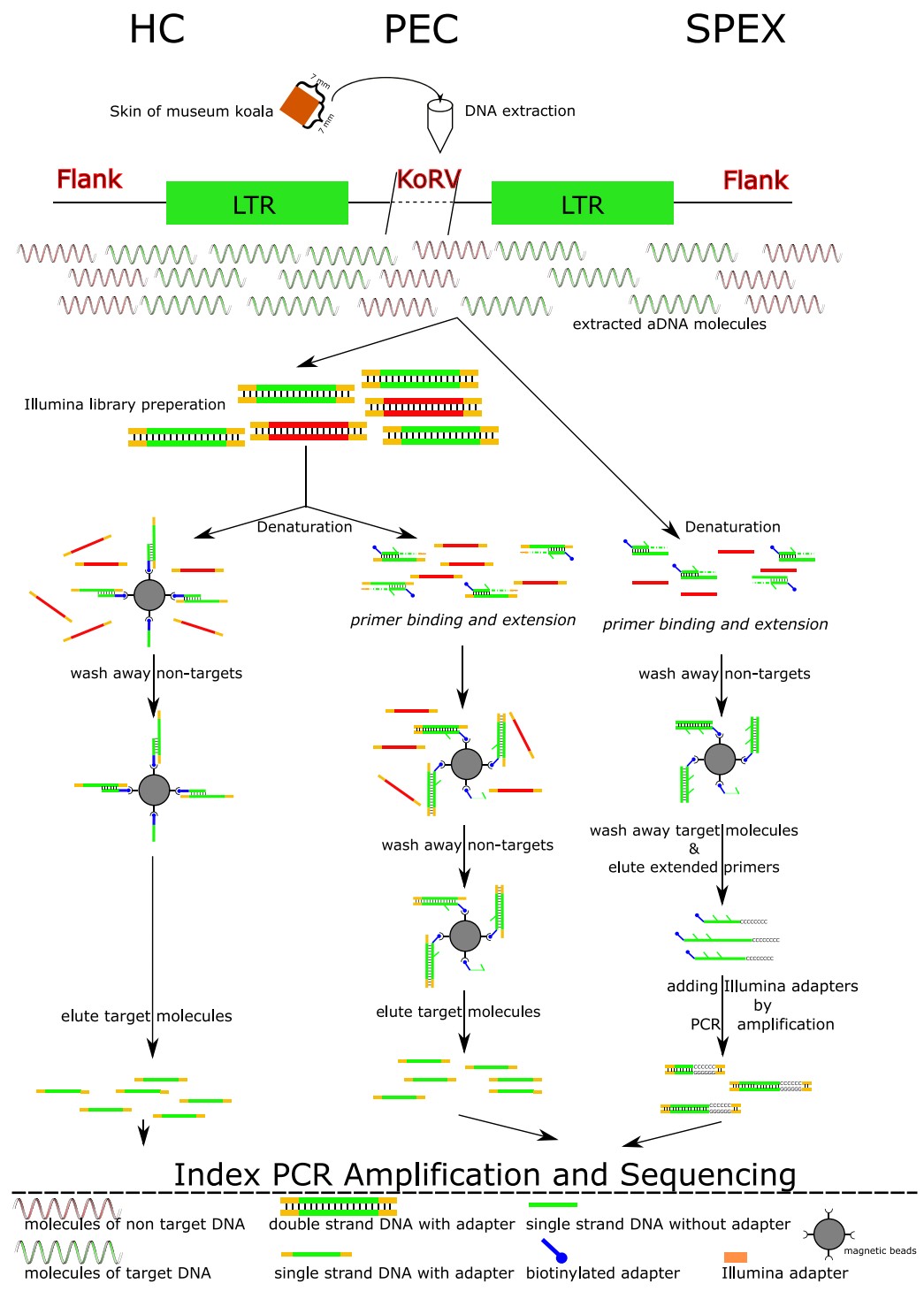

**Figure 1  Experimental work flow for the three enrichment techniques.** Abbreviations: HC, Hybridization Capture; PEC, Primer Extension Capture; SPEX, Single Primer Extension. A square of 7 mm × 7 mm of koala skin tissue per museum specimen was extracted in a dedicated ancient DNA (aDNA) facility. A workflow for the three techniques is illustrated. Both HC and PEC require Illumina library preparation as a preliminary step. The double stranded libraries are denatured to single 
**Figure 1 (…continued)**
stranded DNA molecules and underwent different experimental procedures in HC and PEC. In HC, single stranded DNA libraries are mixed with magnetic beads immobilized with baits. These are incubated by slow rotation at 65 °C for 48 h. After a series of wash steps, the libraries with non-targets sequences are washed off leaving only the libraries with target sequences hybridized with the baits on beads. These target molecules were then dissociated from the baits using a special elution buffer, and were used as templates for PCR amplification. While in PEC, the singled stranded libraries are mixed with biotinylated oligos for 1 min at 55 °C in which only the libraries with target sequences hybridized with the biotinylated oligos. Primer extension reactions of the biotinylated oligos were performed only to these hybridized libraries. Biotinylated oligos were collected by magnetic beads together with the hybridized targeted libraries. The single stranded libraries with target sequences were dissociated with biotinylated oligos and were eluted for subsequent PCR amplification. In contrast for SPEX, DNA extracts are directly denatured to be single stranded and mixed with the same biotinylated oligos used in PEC for 1 min at 55 °C. Similar as in PEC, primer extension reactions of the biotinylated oligos were performed only to the single molecules (target sequences) hybridized with biotinylated oligos. These hybrid molecules were collected using magnetic beads. The original single stranded target molecules were washed away and the biotinylated oligos with 3′ extension were eluted off the beads and were treated with a poly C tailing reaction. These poly C tailed molecules were amplified using primers with a 5′ overhang of the Illumina sequencing adaptor. Through this process, the SPEX products were constructed into Illumina libraries without an additional library preparation step. These SPEX-Illumina libraries were then used in an index PCR and a further amplification step. As shown, SPEX requires at least one more amplification step than HC or PEC, which may explain the high level of clonality in the SPEX result.

work performed in the facility follows procedures designed to minimize the possibility of contamination, such as the use of laminar flow hoods and use of protective clothing. The samples used in this study were all derived from museum skin samples and thus no living koalas were sampled at any point during this study.

DNA from approximately 250 mg of koala skin tissue (7 mm × 7 mm) per museum specimen was obtained using a silica-based extraction kit for aDNA (GENECLEAN Ancient DNA Extraction Kit, MP Biomedicals, USA). The protocol followed the manufacturer's instructions and has been successfully applied to a variety of ancient sample types (*Wyatt et al., 2008*; *Roca et al., 2009*). Mock extractions were performed with each set of koala museum specimens as negative controls for extraction. Subsequent to each extraction, the isolated DNA was further purified using a MinElute spin column (Qiagen, Hilden, Germany) as described in (*Gilbert et al., 2007*) to remove potential inhibitors for the subsequent enzymatic reactions. DNA extracts were not quantified because of the small proportion of endogenous DNA compared to exogenous DNA (contaminants such as bacteria, fungi) in typical aDNA samples.

## NGS Library preparation

Illumina sequencing libraries were prepared from the extracts using a previously described protocol (*Meyer & Kircher, 2010*) with the following modifications: (A) All SPRI purification steps were substituted with spin column purification (MinElute PCR purification kit, Qiagen). (B) Adapter concentration in the ligation reaction was reduced to 0.2 mM per adapter. (C) The purification after adapter fill-in was substituted by heat inactivation at 80 °C for 20 min. The libraries were then used directly as a template for subsequent amplification following a two-step strategy, as previously described (*Kircher, Sawyer & Meyer, 2012*). A quality control strategy (*Meyer et al., 2008*) was also applied,

**Table 1  Koala sample information.**

| Collection no. | Sampling year | Sample provider | Locality | Number in experiment | PCR results[*] |
|---|---|---|---|---|---|
| AMA17300 | 1883 | Australian Museum | New South Wales, Australia; 35 °09′S, 149 °40′E | Koala_1 | Negative |
| AMA17311 | 1883 | Australian Museum | New South Wales, Australia; 35 °09′S, 149 °40′E | Koala_2 | Negative |
| AMA17299 | 1883 | Australian Museum | New South Wales, Australia; 35 °09′S, 149 °40′E | Koala_3 | Negative |
| QM J2377 | 1915 | Queensland Museum | Queensland Australia | Koala_4 | Negative |
| QM J7209 | 1945 | Queensland Museum | Queensland Australia | Koala_5 | Positive |
| QM J8353 | 1952 | Queensland Museum | Queensland Australia | Koala_6 | Negative |
| QM JM1875 | 1960s | Queensland Museum | Queensland Australia | Koala_7 | Positive |
| AM M 12482 | 1971 | Australian Museum | New South Wales, Australia; 33 °38′S, 151 °20′E | Koala_8 | Positive |
| QM JM64 | 1973 | Queensland Museum | Queensland Australia | Koala_9 | Positive |
| QM 7625 | 1970–1980s | Queensland Museum | Queensland Australia | Koala_10 | Positive |
| MCZ 8574 | 1904 | Museum of Comparative Zoology | Queensland Australia | Not sequenced | Poorly working |
| 9111010180 | 1891 | Royal Ontario Museum | Queensland Australia | Not sequenced | Negative |
| 122553 | 1966 | U of Mich Museum of Zoology | Queensland Australia | Not sequenced | Negative |

**Notes.**

[*]PCR results for these samples were reported in *Ávila-Arcos et al. (2013)*.

which consisted of qPCR to quantify the product after each step of library amplification. Based on qPCR results, three samples for which DNA quality was too poor for analysis were excluded from further processing.

In the first round of amplification, AmpliTaq Gold, a non-proof reading enzyme, and indexing primers (Table S1) were applied, adding a distinct P7 index to each library as described in *Meyer & Kircher (2010)*, 10 indices for the 10 working samples and 3 and 4 negative control indices for PEC and SPEX respectively. Adding distinct indices to each library allows for multiple samples to be sequenced in a single sequencing run. The non-proof reading enzyme allows for amplification to be performed on templates containing deoxyuracils, which are common with aDNA (*Der Sarkissian et al., 2015*). After removal of 1 μL for qPCR as a library quality control, the libraries were used as template in 100 μL PCR containing 1x Taq buffer II (Applied Biosystems), 5U AmpliTaq Gold (Applied Biosystems), 250 mM each dNTP and 100 nM each indexing primer. Cycling conditions followed manufacturer's instructions: the pre-denaturation step lasted 12 min at 95 °C, followed by 12 cycles of denaturation at 95 °C for 20 s, annealing at 60 °C for 30 s and elongation at 72 °C for 40 s, with a final extension step of 72 °C for 5 min. PCR products were purified using the QIAquick PCR purification kit (Qiagen, Hilden, Germany).

In the second round of amplification, 5 μL of the purified PCR product from the first round PCR was used as a template for a second PCR. This involved 50 μL reactions containing Herculase II Fusion DNA Polymerase (Agilent Technologies Catalog 600677), which has proof reading activity, and primers IS5 and IS6 (*Meyer & Kircher, 2010*) at a final

concentration of 400 nM each. Cycling conditions included an activation step of 3 min at 95 °C, followed by 15–20 cycles of denaturation at 95 °C for 20 s, annealing at 60 °C for 25 s and elongation at 72 °C for 30 s, with a final extension step at 72 °C for 3 min. The number of cycles used in the PCR for every sample was dependent on the concentration of each of the libraries as determined by the qPCR assay. The PCR amplified libraries were then purified using the QIAquick PCR purification kit. Each library was separately used in subsequent PEC and hybridization capture experiments.

## Bait preparation and integration site enrichment

Three methods were compared for retrieving integration sites: single primer extension (SPEX) primer extension capture (PEC), and hybridization capture. All three have been successfully applied to ancient and historical DNA samples and all are applicable to samples that would not be expected to yield results with conventional methods for integration site analysis. The same set of primers was used in PEC and SPEX experiments (Fig. 2, Table S2). Because the two LTRs of a provirus are identical, the primers designed for enriching the 5′ integrations will also target the 3′ LTR retroviral *env* gene and the primers designed for targeting the 3′ integrations will also extend targeting the retroviral *gag* leader sequence (Fig. 2A). For both the 5′ and 3′ KoRV LTR, two 20 bp primers were developed which overlap such that the 3′ end of the first primer overlapped 8 bp with the 5′ end of the second primer (Fig. 2B, primers 5.1 and 5.2 and 3.1 and 3.2 respectively). To avoid known LTR polymorphisms among KoRV proviruses, the two primers on each side of the LTR were located 17 bp from the 5′ end and 50 bp from the 3′ end of the LTRs in conserved regions (Fig. 2B). The baits used for hybridization capture were synthesized to generate 32 bp oligonucleotides that spanned the full length of sequence covered by primers 5.1 and 5.2 (32 bp) on the 5′ LTR and primers 3.1 and 3.2 (32 bp) on the 3′ end.

## Primer Extension Capture (PEC)

Indexed libraries were pooled in equimolar ratios for primer extension following a published protocol (*Briggs et al., 2009*). After each step, 1 μL of the product was quantified by qPCR. To minimize the amplification bias, each of the captured products was amplified in triplicate, using 5 μL of the captured product as template for each reaction, using the same kit and cycling conditions as described previously under NGS library preparation for second round amplification of Illumina indexed libraries, except that we ran 20 cycles of amplification for all samples. Amplified captured libraries were purified using the QIAquick PCR purification kit (Qiagen, Hilden, Germany) and eluted in 50 μL of elution buffer (EB) and used as a template for a second round of PEC.

## Single Primer Extension (SPEX)

The SPEX experiments generally followed a published protocol (*Brotherton et al., 2007*) using DNA extracts prior to Illumina library construction with three modifications: (1) Illumina sequencing adaptors were attached to the 5′ end of the primers used in the first round of partially nested PCR; (2) MyTaq HS Mix (Bioline, BIO-25045) was used instead of Platinum Taq DNA Polymerase High Fidelity in the first round of a partially nested PCR; (3) only one round of a partially nested PCR amplification was performed. The nested

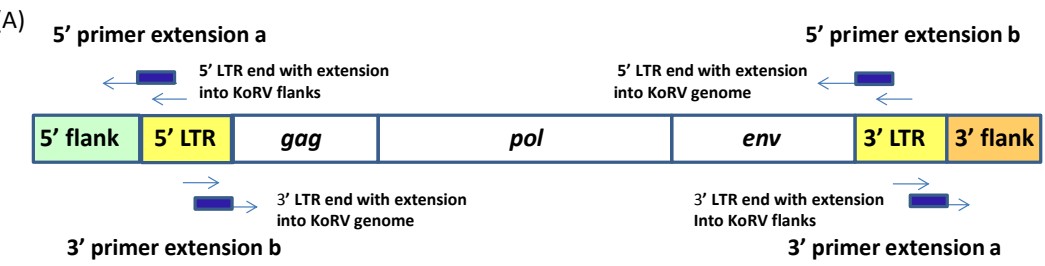

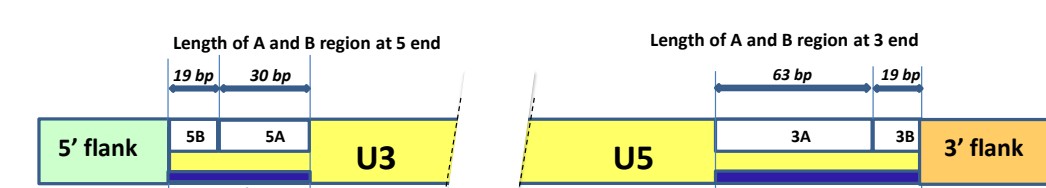

**Genome of koala retrovirus (KoRV) ~ 8.4 kb**

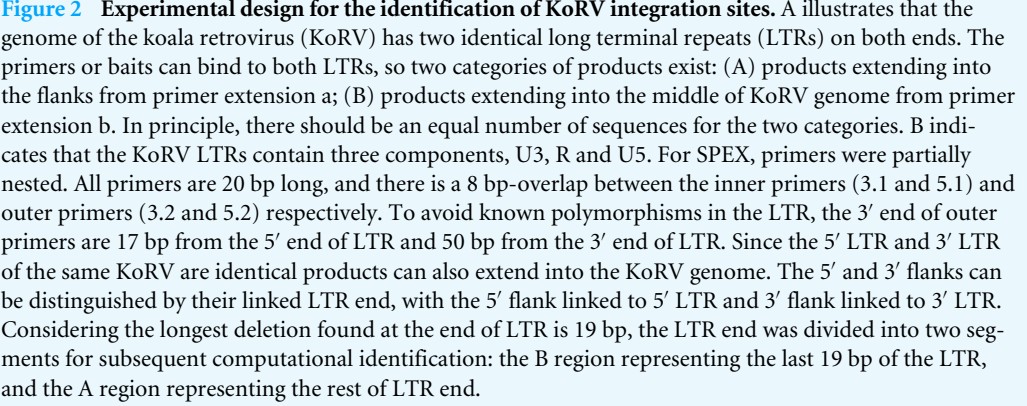

**Figure 2  Experimental design for the identification of KoRV integration sites.** A illustrates that the genome of the koala retrovirus (KoRV) has two identical long terminal repeats (LTRs) on both ends. The primers or baits can bind to both LTRs, so two categories of products exist: (A) products extending into the flanks from primer extension a; (B) products extending into the middle of KoRV genome from primer extension b. In principle, there should be an equal number of sequences for the two categories. B indicates that the KoRV LTRs contain three components, U3, R and U5. For SPEX, primers were partially nested. All primers are 20 bp long, and there is a 8 bp-overlap between the inner primers (3.1 and 5.1) and outer primers (3.2 and 5.2) respectively. To avoid known polymorphisms in the LTR, the 3′ end of outer primers are 17 bp from the 5′ end of LTR and 50 bp from the 3′ end of LTR. Since the 5′ LTR and 3′ LTR of the same KoRV are identical products can also extend into the KoRV genome. The 5′ and 3′ flanks can be distinguished by their linked LTR end, with the 5′ flank linked to 5′ LTR and 3′ flank linked to 3′ LTR. Considering the longest deletion found at the end of LTR is 19 bp, the LTR end was divided into two segments for subsequent computational identification: the B region representing the last 19 bp of the LTR, and the A region representing the rest of LTR end.

PCR products were then quantified by qPCR and indexed using Illumina indexing primers (Table S3). The indexed PCR products were purified using a QIAquick PCR Purification Kit (Qiagen). The amplicons were quantified by qPCR and subjected to a second round of amplification using the same conditions as the first round. The products were purified again using the QIAquick PCR Purification Kit (Qiagen), quantified by qPCR and pooled at equimolar ratios. All PEC and SPEX products were pooled and measured using High Sensitivity DNA chips on an Agilent 2100 Bioanalyzer, then sequenced at the National High-throughput DNA Sequencing Centre, Copenhagen, Denmark using Illumina MiSeq Reagent Kit v2 (300 cycle).

## Hybridization capture

The amplified libraries were pooled in equimolar ratios, totaling a final amount of 2 μg. An established protocol was followed (*Maricic, Whitten & Pääbo, 2010*) except that

synthesized oligonucleotide baits were used instead of PCR products and the EB volume for final elution using Qiagen MinElute column was 20 µL instead of 15 µL. After 2 days of hybridization and subsequent elution steps, 1 µL of the final eluate was quantified by qPCR and 5 µL (in total 15 µL) was amplified in triplicate using the same kit and cycling conditions as described in the NGS library preparation for second round amplification of Illumina indexed libraries. The pooled PCR products were purified using the QIAquick PCR Purification Kit and was measured using the Tapestation 2200 (Agilent Technologies Catalog G2964AA). Hybridization capture libraries were sequenced at the National High-throughput DNA Sequencing Centre, Copenhagen, Denmark using Illumina MiSeq Reagent Kit v2 (300 cycles).

## Preprocessing of sequence data

Adaptor sequences were removed from sequence reads using cutadapt-1.2.1 (*Martin, 2011*), and quality trimming was performed using Trimmomatic-0.22 with default settings (*Bolger, Lohse & Usadel, 2014*). The paired forward and reverse sequence reads were merged using Flash-1.2.5 where possible (*Magoč & Salzberg, 2011*), and both the merged and unmerged reads were used for further analysis. PCR duplicates (clonality in the sequencing data) with 100% sequence identity were removed using cd-hit-v4.6.1 (*Li, Jaroszewski & Godzik, 2001*).

## Identification of KoRV integration sites

Figure 3 and Table S6 summarize the computational pipeline used for the identification of KoRV integration sites. For its implementation, both existing software and customized perl scripts were used that made use of BioPerl (*Stajich et al., 2002*). Because the nested primers or bait were designed near the ends of LTR, the primer extension products would include either the first 49 bp of the 5′ LTR or the last 82 bp of the 3′ LTR, which are designated "LTR ends" in Fig. 2A. All sequences with a KoRV flank should contain an LTR end, as a result of the primer extension (Fig. 2B). Therefore, KoRV integration sites could be identified as the sequence beyond the KoRV LTR end since all integration sites would be attached to an LTR sequence. However, due to DNA degradation in museum samples, some primer extension products may not have a complete LTR end. Furthermore, minor deletions at the end of the integrated LTRs may be present (*Fields, Knipe & Howley, 1996*); for example, a 19 bp deletion was found in a KoRV provirus (*Ishida et al., 2015*). To get around these potential issues, identification of the LTR ends relied on sequentially selecting sample sequences that contain defined LTR segments; this was done in separate steps for the 5′ and 3′ flank-containing sequences. The LTR end was divided into two segments, designated A and B (Fig. 2B): the B segment corresponds to the last 19 bp of the LTR and is referred to as 5B or 3B in the 5′ and 3′ LTR ends, respectively. The A segment is the remaining section of the LTR end, which has a length of 30 bp in the 5′ end (5A) and 63 bp in the 3′ end (3A).

Initially, sequences containing either of the two A regions in the KoRV LTR end (5A or 3A in Fig. 2B) were identified. For this step, optimal local pairwise sequence alignments (Smith-Waterman, EMBOSS *Rice, Longden & Bleasby, 2000*) were computed between each sample sequence and the A region in either the 5′ or 3′ LTR end. Sequences were used for

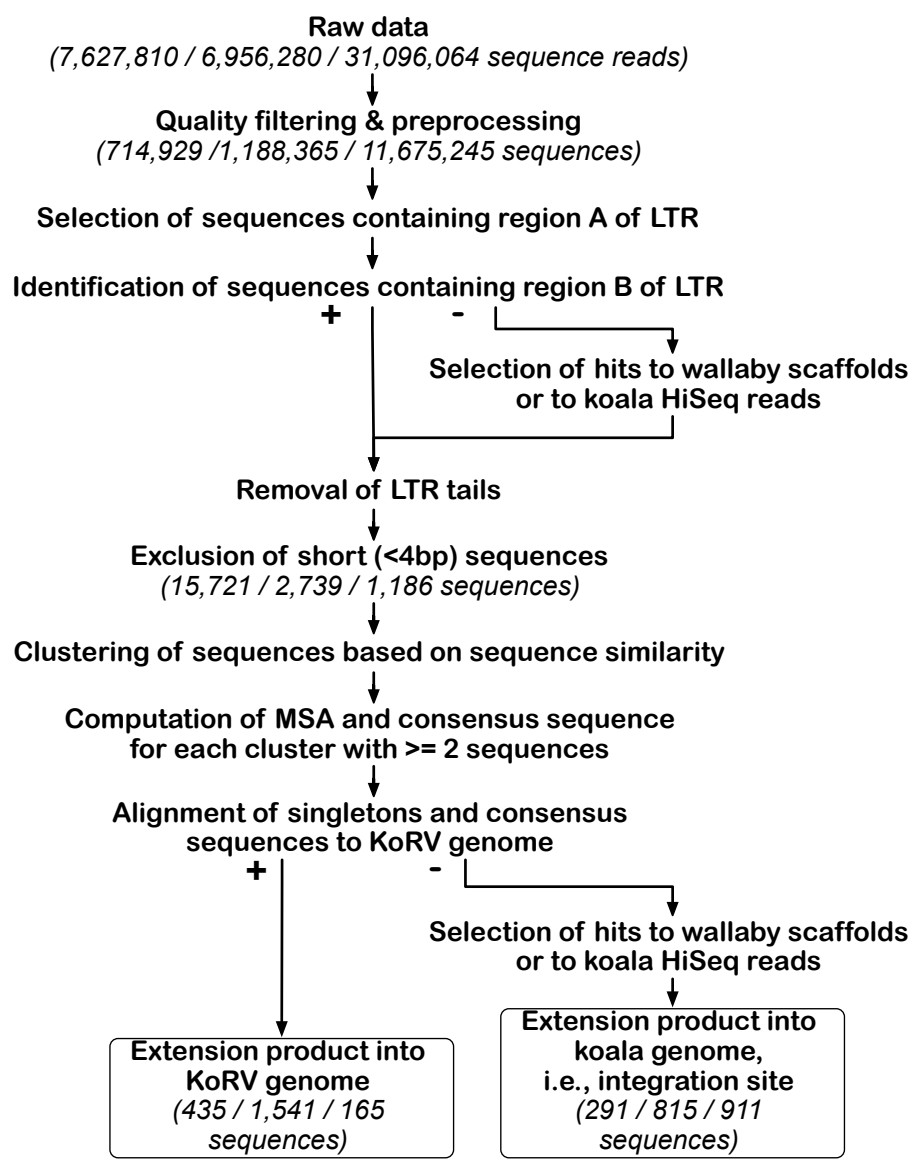

**Figure 3  Bioinformatic pipeline for identification of KoRV integration sites.** The pipeline was run separately for each data set obtained by three different techniques. For the key steps, the number of sequences retained is indicated in parentheses for each technique in this order from left to right: PEC, SPEX and hybridization capture. After processing NGS reads, KoRV integration sites were identified in a two-step analysis of KoRV LTR ends, next to the host DNA flanking KoRV. The first round of selection targeted the A region of the LTR end and its output, was used for subsequent identification of the B region. The LTR ends of all sequences were trimmed off, and only sequences longer than four bp were considered. Using a sequence clustering approach, unique vs. shared integration sites were sorted into clusters. The consensus of each non-singleton cluster was computed using a multiple sequence alignment. These consensus sequences and singleton sequences were queried against wallaby genomic scaffolds and koala Illumina Hiseq reads to determine whether they represented KoRV flanking sequences. At the same time extension products into the KoRV genome were identified.

**Table 2** Selection criteria for 2 rounds of pairwise alignment.

| Step of Filtering | 1st pairwise alignment | | 2nd pairwise alignment | |
|---|---|---|---|---|
| Segment of LTR tail used for alignment | 5A | 3A | 5B | 3B |
| Length of the LTR tail segment in bp | 30 | 63 | 19 | 19 |
| Minimum alignment length in bp | 20 | 43 | 12 | 12 |
| Minimum identity level in percent | 90 | 90 | 80 | 80 |

further analysis if they could be aligned to at least 20 bp of the 30 bp 5A segment with at least 90% identity, or if they could be aligned to at least 43 of the 63 bp 3A segment with at least 90% identity (Table 2). Despite the differences in the lengths of the 5′ and 3′ A segments, this alignment criteria was selected as it resulted in approximately the same sequence identity threshold (∼60%) for both ends. Sequences not passing these criteria were discarded as artifacts. The LTR ends of all sequences meeting these criteria were trimmed to the distal 19 bp and then used for further analysis. A higher sequence identity threshold was not chosen due to potential DNAse degradation of the molecules or ancient DNA based damage lowering homology. A 20 bp sequence length was the minimum on the 5A segment that allowed for LTR identification, whereas the 3A region was longer allowing for a longer minimum segment.

From these sequences, B segments of either 3′ or 5′ LTR ends were identified (3B or 5B in Fig. 2B). For this step, optimal local sequence alignments were computed between each of the trimmed sequence and the B segment in either the 3′ or the 5′ LTR end. Only sequences that could be aligned to at least 12 bp of the 19 bp long B segment (3B or 5B) with at least 80% (Table 2) identity were selected. This criteria was chosen by considering the known polymorphisms in this region of KoRV that originate from the mutagenic properties associated with LTR/host flanking region junctions. The last 19 bp of LTR ends were trimmed from all sequences meeting the selection criteria, leaving LTR free KoRV flanks or KoRV genomic DNA adjacent to the LTR.

All sequences that contained the A region, but for which the B region was not detected using the pairwise alignment strategy, were then subjected to another test. Specifically, these sequences were used as queries for two separate local database searches using BLAST (Altschul et al., 1990). Such sequences represent LTRs that have suffered deletions at the end, a common occurrence in proviruses. One search was against HiSeq sequencing data of a koala from Queensland, Australia with 100X coverage. The data represent raw Illumina sequences and are not annotated or assembled. After adaptor and quality trimming, 6.469 billion reads from this koala, with a mean length of 78 bp, were used for this step. Sequences were considered KoRV integration sites when their non-LTR portion could be aligned with greater than 90% identity to the koala reads over 60% length of the sample sequence. A second search was against the Tammar wallaby (*Macropus eugenii*) genome (GenBank: ABQO000000000.2), which represents the closest related species to koala for which a genome has been assembled (Renfree et al., 2011). Although the wallaby and koala lineages diverged more than 50 Mya (Meredith, Westerman & Springer, 2009), we expected that some of the koala genomic DNA (flanking KoRV) could be aligned to the

**Table 3  Result of analysis for the three technique groups.**

| Technique | SPEX | | PEC | | Hybridization capture | |
|---|---|---|---|---|---|---|
| KoRV flanks orientation | 5 end | 3 end | 5 end | 3 end | 5 end | 3 end |
| KoRV flanks < 4 bp | 15,822 | 1,527 | 496 | 1,806 | 191 | 41 |
| KoRV flanks 4–14 | 6,426 | 8,896 | 329 | 2,033 | 1,052 | 24 |
| KoRV flanks 15 bp or longer | 95 | 304 | 63 | 314 | 106 | 4 |
| KoRV flanks > 4 bp | 6,521 | 9,200 | 392 | 2,347 | 1,158 | 28 |
| Unique insertion sites after clustering | 66 | 182 | 126 | 538 | 862 | 24 |
| Shared insertion sites after clustering | 15 | 28 | 17 | 134 | 25 | 0 |
| Internal KoRV reads | 212 | 223 | 141 | 1,406 | 151 | 14 |
| Total target enrichment products identified | 22,542 | 10,950 | 1,029 | 5,559 | 1,495 | 83 |
| Total sequences after PCR duplicate removal[*] | 714,929 | | 1,188,365 | | 11,675,245 | |
| Efficiency of target enrichment (%) | 4.68 | | 0.55 | | 0.01 | |
| Number of hits to wallaby genome by blast | 1,617 | | 136,366 | | 1,915,781 | |
| Estimated ratio of off-target enrichment (%) | 0.23 | | 11.48 | | 16.41 | |

**Notes.**
[*]The total number of sequences after PCR duplicate removal equals the total number of sequences before pairwise alignment.

homologous wallaby regions. Sequences with at least 70% identity over 50% length of the sample sequence to the wallaby genome were therefore considered to be KoRV integration sites. An assumption made is that KoRV does not frequently or preferentially insert into repetitive sites which could cause us to underestimate the total number of integrations. This will only be resolvable once an annotated koala genome becomes available. For the sequences with a match to the wallaby scaffolds or the koala data, the LTR sequences were trimmed and were then concatenated with the KoRV flanks (obtained in previous steps) for further analysis.

## Sorting of sequences representing different integration sites

All sequences with matches to the different segments of the 3′ and 5′ LTR ends and/or to wallaby scaffolds or koala HiSeq data from each of the enrichment techniques were collected. The sequences matching 3′ and 5′ LTR ends were kept separate, resulting in a total of six different data sets for further analysis (two data sets each for the PEC, SPEX and hybridization capture). LTR ends were removed from all sequences in these data sets. Before using these sequences to identify shared and unique integration sites, all KoRV flanks were sorted into three categories by length (Table 3): (1) including KoRV flank sequences shorter than 4 bp, the typical length of a KoRV target site duplication. These sequences were valid and in the right extension direction but too short for any biological interpretation, and thus were excluded from further analysis. (2) KoRV flanks with length of 4–14 bp. (3) KoRV flank sequences with length of 15 bp or longer. Both KoRV flanks 4–14 bp or with length of 15 bp or greater were used for identifying shared and unique integration sites, but only KoRV flanks of minimum length of 15 bp were used for subsequent pairing of 5′ and 3′ integration sites to one KoRV provirus. Additionally, as per the experimental design (Fig. 2A), each of the 5′ and 3′ primer extension products has two directions of extension, a and b. Extension a is towards the KoRV flanks yielding integration sites as expected, while
**Table 4  Optimized parameters for clustering.**

| Method | PEC | | SPEX | | Hybridization capture | |
|---|---|---|---|---|---|---|
| Insertion site orientation | 5′ end | 3′ end | 5′ end | 3′ end | 5′ end | 3′ end |
| E-value for all versus all blast | 1.00E−17 | 1.00E−20 | 1.00E−30 | 1.00E−30 | 1.00E−15 | 1.00E−15 |
| Inflation value for clustering | 22 | 4 | 1.4 | 4 | 6 | 16 |

extension b is towards the KoRV proviral genome yielding unwanted products for this integration site study. These sequences were designated as *'internal KoRV reads'*. However, despite not representing integration sites, extension b products still represent correctly enriched products from the specific enrichment technique.

Combining the KoRV flank types 4–14 bp long and 15 bp or longer, the PEC data had 392 5′ flank sequences and 2,347 3′ flank sequences; the SPEX data 6,521 5′ flank sequences and 9,200 3′ flank sequences; and hybridization capture 1,158 5′ flank sequences and 28 3′ flank sequences. A clustering approach was used to sort all sequences in each of the six data sets into groups of similar sequences; each cluster representing a unique integration site. Sequences that did not share significant similarity with any other sequences in the input file were called singletons. For each of the six data sets, all-against-all BLAST comparisons were run, and the BLAST output was used as input for clustering using TRIBE-MCL (*Enright, Van Dongen & Ouzounis, 2002*), separately for each data set. Different combinations of E-values (all against all BLAST) and inflation values (TRIBE-MCL) were used for this step and the optimal parameter combination for each data set was evaluated. For all combinations of E-values and inflation values, multiple sequence alignments were computed for all clusters using MAFFT v7.127b (*Katoh et al., 2002*). To assess the quality of the clustering, alignments of the 30 largest clusters of each clustering result were visualized in jalview (*Waterhouse et al., 2009*) and were verified by eye. An alignment was considered high quality if the total number of mismatches and gaps in every sequence of the alignment was no more than 10% of the sequence length. If all 30 clusters were evaluated to be of high quality, the sequence was further analyzed. The parameter combinations for optimal clustering and related all against all BLAST are listed in Table 4.

Singletons and non-singleton clusters containing sequences derived from a single individual koala were considered to represent unique integration sites. Clusters containing sequences shared by more than one koala were considered to represent shared integration sites (Table S4 and S5). A consensus sequence was computed from the alignment of each sequence cluster. Singletons and consensus sequences were then further evaluated first by computing pairwise alignments between these sequences and the *gag* or *env* part of KoRV genome (Fig. 2A) (GenBank: AF151794.2). The sequences that could be aligned to the KoRV genes with at least 90% identity and of any length were categorized as primer extension or flank capture within the KoRV genome. The LTR sequences at the 5′ and 3′ ends of the KoRV genome are identical or nearly so and therefore 50% of the PCR products should extend into the KoRV genome (Fig. 2A). Sequences that could not be mapped to KoRV genome were potential KoRV integration sites and were evaluated further. For such

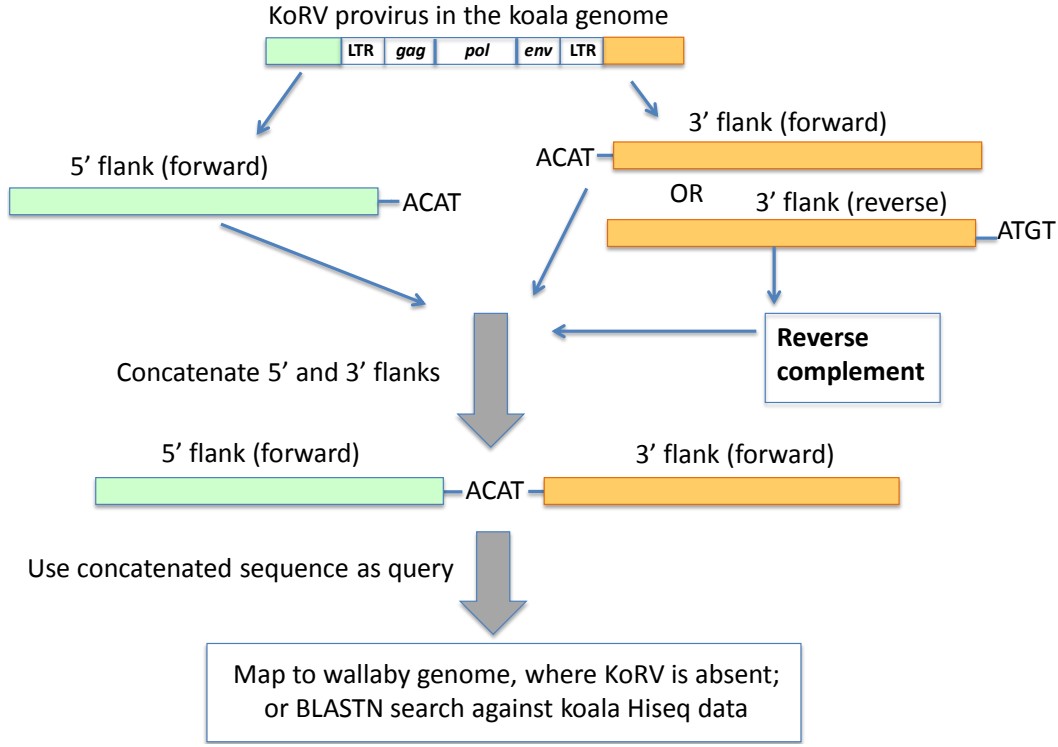

**Figure 4 Pairing of 5′ and 3′ integration sites.** The first 4 bp beyond the KoRV 5′ LTR is the target site duplication (e.g., ACAT in this figure), and the same 4 bp is found at the beginning of a 3′ flank (*Ishida et al., 2015*). One copy of the target site duplication was trimmed off and the two flanks were concatenated. The paired 5′-3′ integration sites were then screened against the wallaby draft genome and koala Hiseq genomic sequences.

sequences, a length filtering was performed with a threshold of 15 bp, since this is the minimum length that can be effectively identified by BLAST. The sequences longer than 15 bp were first used as query in BLAST to search against the koala shotgun Hiseq data; they were also mapped to wallaby genome (GenBank: ABQO000000000.2) in Geneious version 6.18 (http://www.geneious.com, *Kearse et al., 2012*). Identified sequences for either one of the two computations were considered to be KoRV integration sites. Sequences shorter than 15 bp are too short for efficient mapping or BLAST; however, because they contained an LTR end, were included in the KoRV specific enrichment statistics (Table 2), although they were not further analyzed.

## Pairing of 5′ and 3′ integration sites to one KoRV provirus

*Ishida et al. (2015)* identified the length of the retroviral target site duplication (a stretch of host DNA directly adjacent to the retrovirus which is duplicated during retroviral integration) for KoRV to be 4 bp. Based on this target site duplication length (Fig. 4), all 5′ and 3′ integration sites were examined for shared four bp target site identity. Only KoRV flanks 15 bp or longer were used for pairing 5′ and 3′ integration sites. The minimum 26 bp (30 bp minus the four bp target site duplication) combined length discriminated true wallaby matches from non-significant blastn results.

The paired 5′-3′ flanking sequences were (1) mapped against the wallaby genome using the mapping tool in Geneious using default settings, where only the paired 5′-3′ integration sequence that could be mapped to the wallaby genome with over 70% of their total length were scored as positively identified; (2) used as query to search in the HiSeq data of Queensland wild koala using BLAST. Here, only the paired 5′-3′ integration sites that could be aligned with over 90% identity with the koala HiSeq reads were considered positive.

### Statistical analysis of shared integration sites

Statistical tests were performed to check if the occurrences of KoRV at sampled integration sites increased as the samples became younger among the 10 museum koala samples. Two logistic regression models were employed: one for 5′ integration sites and one for 3′ integration sites. Both models had the same structure. The occurrence was considered (binary: 1 = presence, 0 = absence) as the response variable and time as a continuous fixed effect. Because results were qualitatively similar irrespective of expressing "time" as rank or directly as years, for the sake of simplicity, only the latter was reported. The identity of koalas and of insertion sites were considered as two Gaussian random effects, making this logistic regression a Generalised Mixed effect Model (GLMM). The GLMM was fitted using the function HLfit from the R package spaMM 1.4.1 (*Rousset & Ferdy, 2014*), considering a Binomial error structure. The effect of time was tested by performing an asymptotic Likelihood Ratio Test (LRT) using the function anova.HLfit from the same package.

## RESULTS

NGS sequencing post enrichment by all three tested methods generated hundreds of thousands to millions of reads. The reads displayed the typical length distribution of aDNA (Fig. S1). After pre-processing steps, 714,929 sequences from the SPEX approach were available for further analysis, 1,188,365 from PEC, and 11,675,245 from hybridization capture.

### Single primer extension (SPEX)

Using SPEX to target the 5′ LTR flanks, 66 integration sites unique to a single koala, and 15 integration sites shared by at least two koalas were identified across the 10 koala samples, for sample descriptions see Table 1. These integration sites derived from consensus sequences generated from sequence clusters with at least 4 bp of sequence (representing the length of the target site duplication of KoRV) (*Ishida et al., 2015*) flanking the KoRV LTR (categorized as either KoRV flank sequences 4–14 bp long or 15 bp or longer in Table 3). An additional 15,822 sequences were less than 4 bp; these could not be further analyzed since their length was shorter than the target site duplication, these are listed as KoRV flanks shorter than 4 bp in Table 3. Additionally, 212 reads were identified as part of the *envelope* gene of the KoRV genome. This results from the presence of identical primer target sites in the 5′ and 3′ LTRs (Fig. 2A), since KoRV 5′ and 3′ LTRs are identical or nearly so (*Ishida et al., 2015*). Thus, approximately 50% of the sequences are expected to be internal KoRV proviral reads that extend from the LTR into the proviral genome rather than into the host flanking region. For clarity, we term these sequences '*internal KoRV reads*'. These

sequences that extended into the KoRV genome were categorized separately but included in the total enrichment efficiency evaluation because they still represent correct enrichment of target sequences. SPEX for integration sites next the 3′ LTR also identified 182 unique and 28 shared 3′ LTR flanks; with 1,527 sequences being too short for further analysis (less than 4 bp of flank sequence) and 223 internal KoRV reads that matched the KoRV genome (Table 3).

### Primer extension capture (PEC)

PEC was designed to identify flanking regions 5′ of integration sites and detected 126 unique and 17 shared integration sites. An additional 496 sequences included less than 4 bp of flank that was too short for further analysis, while 141 internal KoRV reads extended into the KoRV genome. PEC targeting regions downstream of the 3′ LTR integration sites identified 538 unique and 134 shared integration sites. An additional 1,806 reads were less than 4 bp, while 1 internal KoRV read was identified that matched KoRV (Table 3).

### Hybridization capture

Using the 5′ LTR region as bait, 862 unique and 25 shared 5′ flanking regions were identified by hybridization capture. An additional 191 sequences included less than 4 bp of flank and 151 internal KoRV reads were characterized. Using the 3′ LTR region as bait, only 24 unique and no shared integration sites were identified by hybridization capture. The strong bias of this method towards identifying 5′ integration sites has been previously observed (*Tsangaras et al., 2014b*). Additionally, 41 sequences included less than 4 bp of flank, while 14 sequences were classified as internal KoRV reads (Table 3).

### Summary of computational data processing

At each step of our bioinformatics pipeline, we recorded for each experiment the number of sequences that met our screening criteria (Fig. 3). The mean length, minimum length and maximum length of sequences were also calculated at each step (Table S6). Before any screening criteria were applied, SPEX produced 7,628 million, PEC produced 6,956 million reads, and hybridization capture produced 31,096 million. After screening and PCR duplicate removal of this sequencing data, 9.37% of the initial sequencing reads were kept for SPEX, 17.08% for PEC, and 37.55% for hybridization capture. Clonal sequences i.e., duplicate sequences resulting from PCR bias in amplification were more prevalent for products of SPEX than for products of either PEC or hybridization capture.

Two rounds of bioinformatics LTR end identification was performed (Fig. 3). After the first round of LTR end identification, 142,577 (19.94% of the reads after pre-processing) LTR positive sequences were identified for SPEX, 31,787 (2.67% of the reads after pre-processing) for PEC, and 5,648 (0.05% of the reads after pre-processing) for hybridization capture. Sequences passing the second round of 5′ LTR end selection were 22,542 for SPEX, 1,029 for PEC, and 1,495 for hybridization capture, while the sequences passing the second round of 3′ LTR end selection were 10,950 for SPEX, 5,559 for PEC, and 83 for hybridization capture. No KoRV LTR ends were detected in negative controls, extraction or PCR controls lacking template, for any experiment.

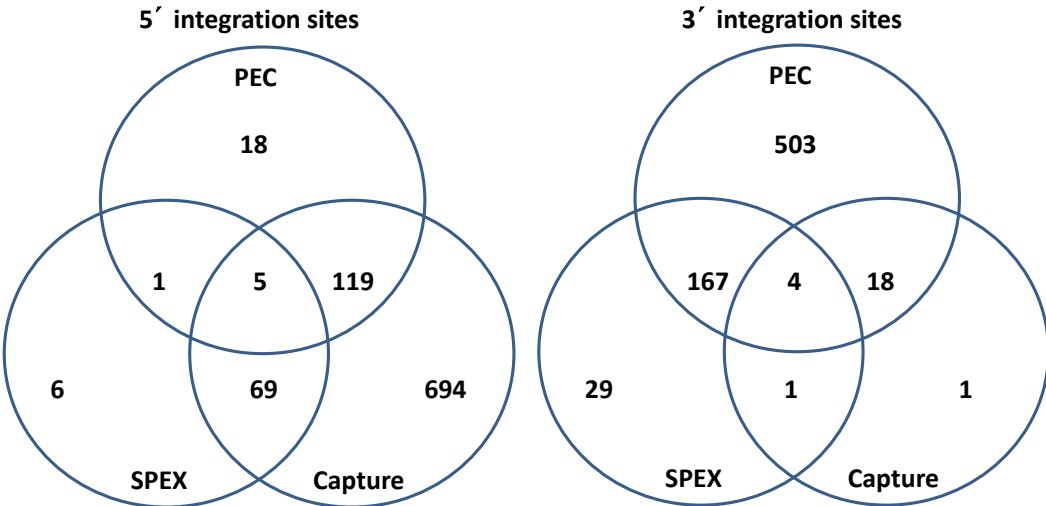

**Figure 5  Venn diagrams of KoRV integration sites found by different methods.** (A) For 5′ integration sites, HC (hybridization capture) yielded the highest total number of integration sites (887), and covered 91.3% of the integration sites found by SPEX and 86.7% of the integration sites found by PEC. (B) For 3′ integration sites , PEC yielded the highest total number of integration sites (672), and covered 81.4% of the integration sites found by SPEX and 91.7% of the integration sites found by hybridization capture. For the retrieval of both 5′ and 3′ integration sites, SPEX showed the worst performance (smallest number of integration sites retrieved among three enrichment methods).

## Cross-technique comparisons

The efficiency of target enrichment for each technique was calculated as the total number of identified flank sequences divided by the total number of sequences after removal of clonal sequences. The total number of target enrichment products included KoRV flanking sequences of any length and internal KoRV reads.

As shown in Table 3, PEC enriched the highest total number of 3′ integration sites, 672, whereas hybridization capture enriched the most 5′ integration sites, 887. As a percentage of the total sequences retrieved, SPEX achieved the highest target enrichment efficiency (4.68%). Both PEC and hybridization capture exhibited lower enrichment percentages (0.55% and 0.01% respectively).

Due to a phenomenon known as CapFlank (*Tsangaras et al., 2014a*), koala genome sequences near the integration sites may be enriched together with KoRV flanks by concatenation of library molecules on the baits. To estimate the numbers of such target flanks, after PCR clonal sequence removal, all sequences were screened using BLAST against the wallaby, which represents the phylogenetically closest species to the koala with an assembled genome. Hybridization capture exhibited the lowest efficiency of on-target enrichment (0.01%, Table 3) and highest ratio of CapFlank enrichment (16.41%), while SPEX achieved the highest efficiency of on-target enrichment (4.68%) and lowest ratio of CapFlank enrichment (0.23%).

As illustrated in Fig. 5, for the 5′ LTR integration sites, hybridization capture yielded the highest total number of integration sites, 887, and contained 91.36% of the integration sites identified in the SPEX data set and 86.71% of the integration sites identified in PEC

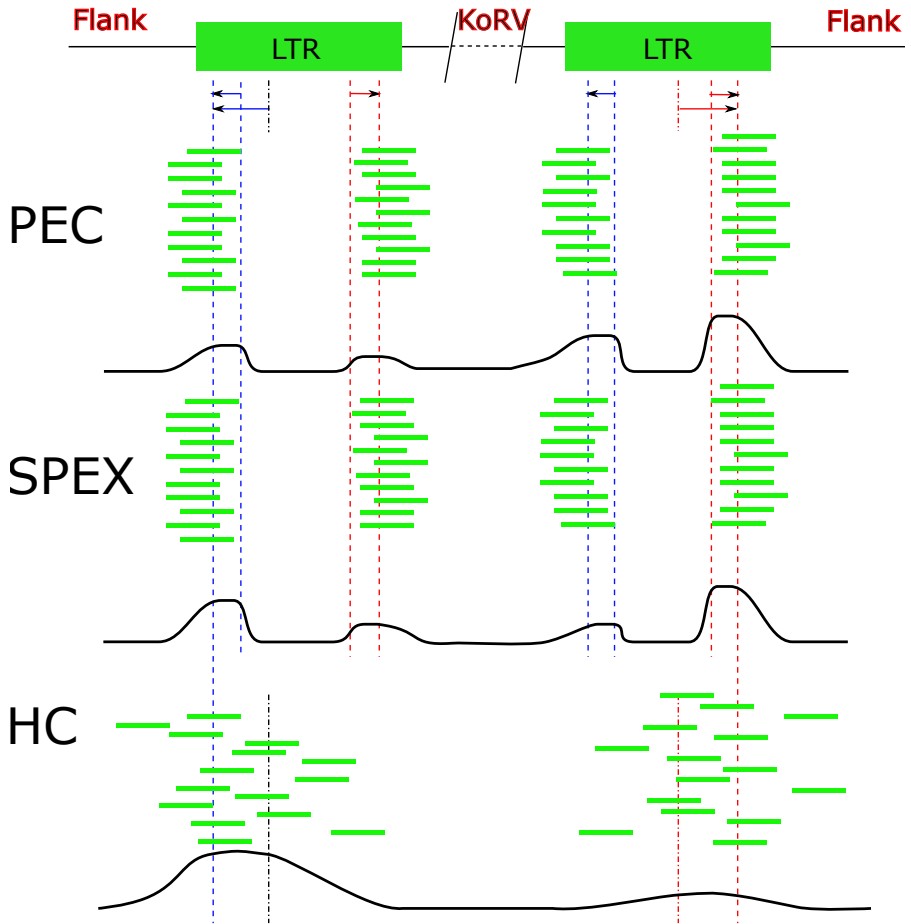

**Figure 6** **Target sequences distribution of three techniques.** Abbreviations: HC, Hybridization Capture; PEC, Primer Extension Capture; SPEX, Single Primer Extension. The oligos used for all three experiments bind near to the end region of the KoRV LTR. Because the genome of KoRV has two identical long terminal repeats (LTRs) on both ends, primer extension of captured products using these oligos will yield two categories of products; (i) KoRV flanks, the desired products for this study which extend into the koala DNA flanking KoRV and (ii) Internal KoRV reads, sequences extending towards the middle of KoRV genome. The bold black line at the bottom of each technical section approximately present the target sequences in the final result, showing a bias towards the 3′ end for PEC and a bias towards 5′ for HC.

data set. The 3′ LTR integration data followed a different profile with PEC generating the highest total number of integration sites, 692, containing 85.07% of the integration sites in the SPEX data set and 91.67% of the integration sites in the hybridization capture data set. The expected enrichment profile and approximate location of the recovered reads based on each of the three methods performance is shown in Fig. 6.

## Shared and unique integration sites

After identical integration sites across the data sets generated by the 3 techniques were combined, 52 shared and 865 unique 5′ KoRV host flanks could be identified. Shared integration sites accounted for 5.7% of the total number identified using 5′ flanking host sequences, a similar percentage as estimated in a previous study (*Tsangaras et al., 2014b*).

Among the 3′ flanking regions, 146 shared and 570 unique integration sites were identified, with shared sites accounting for 20.4% of total integration sites identified using 3′ host genomic sequences.

## Pairing of 5′ and 3′ flanking regions to identify individual proviral integration sites

KoRV typically produces a 4 bp target site duplication upstream and downstream of its integration site (*Ishida et al., 2015*). All 4 bp putative 5′ target site duplications were compared to all 4 bp putative 3′ duplications. In cases where there was an exact match at the 4 bp, the two flanking regions were concatenated to simulate the sequence that a virus-free host would have at that locus, assuming that the target site duplications actually were from the same locus. A total of 1,690 concatenated 5′ and 3′ host flanking sequences were used to query the koala HiSeq data to identify proviral integration sites (Fig. 4). There were 63 matches, indicating that the 5′ and 3′ flanks actually corresponded to integration sites at the same KoRV proviral locus. Of these 63 loci, 40 corresponded to proviral integration sites present in a single koala (Data S1), whereas 23 corresponded to a proviral integration site detected in at least two koalas (Data S2).

## The comparison of integration sites across different studies

The KoRV integration sites identified by this study were compared to those reported by *Tsangaras et al. (2014b)* and *Ishida et al. (2015)* (Table S3). Each study used a different set of koalas, and there was no overlap in koala individuals examined by the three studies. All but one of the koala specimens used by *Tsangaras et al. (2014a)* and *Tsangaras et al. (2014b)* and compared to our results were museum samples. By contrast, all the koalas examined in *Ishida et al. (2015)* were from modern samples.

For the 3′ integration sites, no sharing of integration sites between the museum samples in this study and museum samples in *Tsangaras et al. (2014b)* was detected. Two integration sites were found to be shared between the two youngest museum samples of the current study and Pci-SN265 (the only modern koala in *Tsangaras et al. (2014b)*). Moreover, one integration site was found shared between a modern koala (Pci-SN248) in *Ishida et al. (2015)* and Pci-SN265 of *Tsangaras et al. (2014b)*. One integration site was also found shared between two museum koalas in the current study and modern koalas in *Ishida et al. (2015)* (Table S4).

Among 5′ integration sites, three were shared between the museum samples in this study and those used in *Tsangaras et al. (2014b)*. Two integration sites were found to be shared between the museum samples of this study and Pci-SN265, and two integration sites were found shared between modern koalas (including Pci-SN248) in *Ishida et al. (2015)* and Pci-SN265. Additionally, four integration sites were found shared between relatively young museum koalas in this study and modern koalas in *Ishida et al. (2015)*. A 5′ integration site (KoRV-5-shared_7) was shared by 9 koalas, including 4 museum koalas in this study, 4 museum koalas in *Tsangaras et al. (2014b)*, and one modern koala (Pci-SN404) in *Ishida et al. (2015)* (Table S5). Statistical modeling of shared KoRV integration sites among 10 koalas showed an increased sharing of integration sites over time. The details are described in Article S1.

## DISCUSSION

The currently available software for identifying viral integration sites using NGS data require an assembled host genome as a reference, e.g., SLOPE (*Duncavage et al., 2011*), VirusFinder (*Wang, Jia & Zhao, 2013*) and VirusSeq (*Chen et al., 2013*). For the koala however, no assembled genome is available, only raw sequence reads averaging 98 bp in length. We therefore established a customized computational pipeline that was largely reference-independent, but it made use of the Illumina HiSeq reads of the koala and assembled scaffolds of the wallaby, the closest relative to the koala with an assembled genome (*Renfree et al., 2011*).

Given the typically degraded state of DNA in museum specimens, many of the captured or extended molecules in this study either did not extend beyond the LTR or extended only a few bases into the flank. However, such sequences still represent successfully targeted enrichment even if they did not provide extensive integration site information. Primers closer to the ends of the LTRs may have retrieved more and longer integration site data. However, polymorphisms within the ends of the LTRs (*Ávila-Arcos et al., 2013*) may have led to primer mismatch, reducing the effectiveness of all three methods in identifying integration sites. The distance between the primer target and the end of the 5′ LTR was 37 bp, whereas for the 3′ LTR the distance was 70 bp. This may explain why the sequencing following hybridization capture yielded more 5′ flanking regions than 3′ flanking regions. However, primer position may not be the only factor, since both PEC and SPEX yielded more 3′ integration sites overall even though the primers were identically positioned. The LTRs of KoRV are distinct from those of its known closest related viruses, the gibbon ape leukemia virus (GALV). However, we cannot exclude the possibility that additional KoRV like LTRs exist in the koala genome associated with distinct ERVs that may lead to an overestimate of integration sites. However, over a decade of molecular biological analysis of KoRV like viruses in koalas have not identified such closely related ERVs in any species including koalas.

Both techniques that involve extension from a primer (SPEX and PEC) were biased toward the 3′ integration sites whereas techniques that did not extend from a primer (hybridization capture and genome-walking) were not. The underlying mechanisms generating this bias are not clear. Several koala samples in the current study overlap with those examined by PCR (around 100 bp amplifications) in *Ávila-Arcos et al. (2013)* (Table 1). Several samples in that study failed to yield PCR products but were successful here, likely because shorter sequences, less than 100 bp, are easily retrieved by the methods applied by the current study.

Hybridization capture found the greatest number of 5′ integration sites, which included nearly all integration sites identified by SPEX and 86.71% of the integration sites identified by PEC (Fig. 5). In contrast, for the 3′ LTRs, PEC yielded the most integration sites including 85.07% and 91.67% of the integration sites identified by SPEX and hybridization capture respectively. The results were generally consistent across individuals and with the data pooled (Table S6), with no single sample driving the biases for the 5′ or 3′ integration site retrieval thereby validating the reliability of the methods tested in this study. Considering

the output of the methods, the most reliable and comprehensive screening of museum DNA for sequences flanking a target can be achieved by performing PEC and hybridization capture in combination. Both methods covered nearly the full diversity of integration sites identified by SPEX. However, PEC and hybridization capture each retrieved integration sites unique to the method and had reciprocal biases in retrieving 5′ and 3′ integration sites. It should also be considered that because not all integration sites could be paired for 5′ and 3′ LTRs, it is clear that not all integration sites present in the samples were retrieved, even when combining all methods. The strong biases towards the 5′ or 3′ integration sites may prevent such comprehensive analysis from historical samples except at very high sequence coverage depth, for example, using Illumina HiSeq sequencing.

Querying of sequences concatenated from 5′ and 3′ flank sequences that suggested identical target site duplications identified 63 matches using the wallaby genome as a reference. The success rate would likely improve upon the availability of an assembled koala reference genome. Genome data available to this project was represented by unassembled raw reads of 98 bp average length. Among the 63 KoRV integration sites identified by this method, 40 were identified after concatenating 5′ and 3′ flanks derived from the same individual koala. A total of 23 integration sites were identified by querying with a sequence that concatenated 5′ and 3′ flanking sites from different koala individuals. This result demonstrates that although many integration sites were identified per koala, they were not identified comprehensively and many integration sites were missed. Considering that there are an estimated 165 KoRV copies per haploid genome in Queensland koalas (*Tarlinton, Meers & Young, 2006*), exhaustive identification of integration sites would have required detection of 1,650 5′ and 3′ integration sites across the 10 koalas used in the study. Moreover, for aDNA, comprehensive identification of integration sites is even more challenging due to the poor and variable condition of the samples, which results in a decrease in the number of endogenous DNA copies.

Little sharing of integration sites between museum samples in this study and those in *Tsangaras et al. (2014b)* were found (none at 3′ and three at 5′). This is possibly due to the methodology difference between the two studies: in *Tsangaras et al. (2014b)*, the integration sites and the ends of KoRV LTRs were intentionally avoided for targeted hybridization capture retrieval of KoRV proviral sequences. The integration sites in this same study were captured due to the high CapFlank (*Tsangaras et al., 2014a*) nature of hybridization capture. In contrast, our study specifically focused on targeted retrieval of integration sites, which were more intensively studied using three techniques. *Ishida et al. (2015)* also used a different technical strategy than this study, namely genome walking. The focus of their study, like our own, was also integration site retrieval. This is evidenced by a slightly higher number of integration sites shared between museum koalas of our study and modern koalas in *Ishida et al. (2015)*.

Generally, the low number of shared integration sites between the three studies can be due to the varying level of intensiveness for KoRV flank retrieval, which can potentially miss many shared integration sites. Given the independent aims and methods used across the three studies, statistical modeling of shared KoRV integration sites through time was only performed for the ten museum koalas in this study (Article S1; Fig. S2). While the

number of koalas examined is few, a statistically significant increase in integration site sharing was observed over time. This could be explained by increased fixation of KoRVs over time. However, with only ten samples, regional differences in fixation of KoRVs e.g., the three young koalas from NSW could also explain the trend as a geographic rather than temporal trend. The methods applied in the current study should allow for a broader screen of museum koalas to distinguish between these possibilities. However, the current study confirms that in general, koalas share few integration sites among individuals in Queensland where KoRV is ubiquitous which contrasts with most known ERVs which are either fixed in the genome of the host species or are at very high frequency. This is further evidence that the KoRV invasion of the koala genome is still in the early stages.

## CONCLUSIONS

A combination of PEC and hybridization capture generated the most comprehensive coverage of retroviral integration sites from historical samples. This is consistent with the high coverage of both provirus and integration sites observed in previous hybridization capture studies on modern and historical koalas (*Tsangaras et al., 2014b*). If mapping to an annotated genome were possible, clustering and other bioinformatic analysis would be facilitated. However, without an annotated reference genome, the methods described here allow for thorough characterization of high copy retroviral integrations. KoRV exhibits only a small fraction of shared integration sites among koalas consistent with its recent invasion of the koala genome. The methods described here should facilitate the characterization of target flanking sequences of any kind from modern and historical samples.

## ACKNOWLEDGEMENTS

The authors thank Jörns Fickel for computational support. The authors also thank Joachim Selbig and his staff in the Research Group of Bioinformatics, Institute for Biochemistry and Biology, University of Potsdam. For museum specimens, we thank F Johansson and G Nilson (Bohusläns Museum), R Timm (Natural History Museum—University of Kansas), J Chupasko and H Hoekstra (Harvard Museum of Comparative Zoology), W Longmore (Museum Victoria), O Grönwall and U Johansson (Swedish Natural History Museum), J Eger (Royal Ontario Museum), S Hinshaw (University of Michigan Museum of Zoology), D Stemmer and C Kemper (South Australian Museum), S Ingleby (Australian Museum), S. Van Dyck and H Janetzki (Queensland Museum). The content is solely the responsibility of the authors and does not necessarily represent the official views of the NIGMS or the National Institutes of Health.

### Funding

YI, ALR, KMH and ADG were supported by Grant Number R01GM092706 from the National Institute of General Medical Sciences (NIGMS). ADG was additionally supported by a grant from the Morris Animal Foundation, grant number D14ZO-94. PC was

supported by a fellowship from the China Scholarship Council. UL was supported by the interdisciplinary training initiative "Evolution across Scales", funded by the Volkswagen foundation (Grant Number 83459). DEAP was supported by a scholarship from the Deutscher Akademischer Austauschdienst–DAAD. The funders had no role in study design, data collection and analysis, decision to publish, or preparation of the manuscript.

## Grant Disclosures

The following grant information was disclosed by the authors:
National Institute of General Medical Sciences (NIGMS): R01GM092706.
Morris Animal Foundation: D14ZO-94.
China Scholarship Council.
Volkswagen foundation: 83459.
Deutscher Akademischer Austauschdienst–DAAD.

## Competing Interests

The authors declare there are no competing interests.

## Author Contributions

- Pin Cui conceived and designed the experiments, performed the experiments, wrote the paper, prepared figures and/or tables, reviewed drafts of the paper.
- Ulrike Löber conceived and designed the experiments, analyzed the data, wrote the paper, prepared figures and/or tables, reviewed drafts of the paper.
- David E. Alquezar-Planas wrote the paper, prepared figures and/or tables, reviewed drafts of the paper.
- Yasuko Ishida performed the experiments, reviewed drafts of the paper.
- Alexandre Courtiol and Dorina Lenz analyzed the data, reviewed drafts of the paper.
- Peter Timms, Rebecca N. Johnson and Kristofer M. Helgen contributed reagents/materials/analysis tools, reviewed drafts of the paper.
- Alfred L. Roca wrote the paper, reviewed drafts of the paper.
- Stefanie Hartman and Alex D. Greenwood conceived and designed the experiments, wrote the paper, prepared figures and/or tables, reviewed drafts of the paper.

## Animal Ethics

The following information was supplied relating to ethical approvals (i.e., approving body and any reference numbers):

The samples used in this study were all derived from museum skin samples and thus no living koalas were sampled at any point during this study.

## DNA Deposition

The following information was supplied regarding the deposition of DNA sequences:

The Illumina generated sequences were deposited in the Sequence Read Archive (SRA) for each koala samples (SRR2015712, SRR2016453, SRR2016454, SRR2016455, SRR2016645, SRR2016646, SRR2016647, SRR2016649, SRR2016650, SRR2016651).

## Data Availability

The scripts generated in the study have been provided as a supplemental data file (Data S3). The data that the scripts were applied to have been deposited in the Sequence Read Archive.

## Supplemental Information

Supplemental information for this article can be found online at http://dx.doi.org/10.7717/peerj.1847#supplemental-information.

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
