# Peer review of "Comprehensive profiling of retroviral integration sites using target enrichment methods from historical koala samples without an assembled reference genome"

_PeerJ, doi:10.7717/peerj.1847_

## Round 0.1 · original submission · Minor Revisions

· Academic Editor

Minor Revisions

On balance of the three expert reviews, I consider corrections to fall under minor revisions. Please address all comments where you may disagree with restructuring of the document, please make a case for not doing so.
May i also remind that any code (perl scripts are mentioned) are made publically available prior to publication. This could be as a supplemental file or more appropriately on an external site, e.g github.

Reviewer 1 ·

Basic reporting

LN 55-56: 'Unique opportunity' is not strictly true. EAV-HP in chicken is similar to KoRV in respect to its ongoing endogenization, and vertical transmission of EAV-HP integration sites reflect population structure. The authors may wish to consider drawing some parallels between EAV-HP and KoRV in this regard (e.g. LN 51-56; 74-80), in which case consider the below references, which provide bioinformatic and molecular case studies, respectively.

The methods indicates that these simulated sequences were mapped against the wallaby genome (LN 356) and queried against koala HiSeq data using BLAST (LN 360), whilst the results suggests they were used to query the koala genome (LN 473); the text should ideally be clarified for consistency.

I would suggest the Figure 4 title be changed to 'Pairing of 5' and 3' integration site flanking sequences'. There is, after all, only a single integration site in the host per viral integration, and so pluralising 'site' is perhaps not strictly correct.

The supplementary methods on statistic modelling of shared KoRV integration sites requires formatting for consistency. At present there is a variety of font type and size, and indentation. There is also doubt over the presence/absence of references – as indicated in the document comments.

1. Wragg, D. et al. Genome-wide analysis reveals the extent of EAV-HP integration in domestic chicken. BMC Genomics 16, 784 (2015).
2. Sacco, M. A. & Venugopal, K. Segregation of EAV-HP ancient endogenous retroviruses within the chicken population. J. Virol. 75, 11935–11938 (2001).

Experimental design

LN 185-187: Do the LTRs of any other ERVs share high sequence identity with those of KoRV, which might lead to an over-estimation of KoRV sites? For instance, a BLAST of of the LTR from a KoRV isolate (KJ152817) returns hits for sarcoma virus in the woolly monkey (79% identity over 73% alignment; E = 4-55), leukemia virus in the gibbon ape (79% identity over 69% alignment; E = 5-54), murine leukemia virus in the mouse (75% identity over 56% alignment; E = 2-28). Is there a possibility that some of the sequences obtained are non-specific to KoRV, but may be from other ERV present in the koala? This should perhaps be considered in the discussion, even if only briefly as there may be a lack of characterised ERVs in the koala due to lack of reference genome.

LN 258- 261: Justification for retaining sequences that align to at least 20/30 bp (5A) and 43/63 bp (3A) with 90% similarity, equivalent to > 60% overall sequence identity, could be clearer.
LN 266-268: As above, justification could be made clearer for retaining sequences with > 50% identity (12 * 0.8 / 19).
LN 278-280: Is there a possibility of KoRV insertion into homologous/repetitive elements with the genome? - In which case, would the criteria applied here (90% identity over 60% length) result in over- or under-estimation of integration sites? Similar thoughts apply with regards to orthologous sequence in the wallaby LN 285-287 (70% identity over 50% length).

Validity of the findings

No Comments

Comments for the author

This study provides an interesting analysis of host integration sites of the koala endogenous retrovirus KoRV. As there is no assembled reference genome for the koala the authors developed a computational methods to analyse the results of three target enrichment techniques, with the aim to characterise KoRV integrations in museum specimen. I find the manuscript to be well-written and have no major concerns regarding the authors' conclusions. I have a few additional comments to those provided under 'basic reporting' and 'experimental design'.

Pairing 5' and 3' flanking sequences based on the target site duplication (TSD) and flanking sequence is a good initiative. By concatenating the sequence flanking the TSD, they simulate the host sequence absent of the virus, and can align the same to a reference to identify the integration site. Typically the quality of the reference genome is a limiting factor when mapping virus-host flanking sequences. However in using the wallaby genome, which is devoid of KoRV, the authors appear to side-step this issue. NCBI indicates the wallaby genome to have an N50 size of 2.6 kb and 36.6 kb for contigs (n = 1.17 M) and scaffolds (n = 277 k), respectively. Had a reference genome of comparable quality been available for the koala then it would contain KoRV elements. In theory, a KoRV integration in such a reference genome could be split across across different genomic contigs/scaffolds due to difficulty in uniquely assembling multiple integrations given the length of the virus genome, and the sequence identity of gag, pol and env elements across different viruses. Thus, the 5' and 3' flanking sequences for a given integration site in the reference might be located on different contigs/scaffolds. In such a case, the concatenated flanking sequence approach would be suitable for identifying integration sites absent in the reference, assuming the host sequence flanking the integration was contiguous. Partial alignments arising from the presence of KoRV might then be identified using existing software, as mentioned by the authors (LN 503-505). The authors appear to have generally acknowledge these issues, as indicated in Figure 4 [Map to wallaby genome, where KoRV is absent], however they have not explicitly discussed them in the text.

The cross-technique comparisons, for instance as illustrated in Fig. 5, considers the total data for each target enrichment method. Shared and unique integration sites are likewise compared across the total data generated by the each target enrichment method. The authors may wish to consider also reporting the cross-validation of identified flank sequences by each target enrichment method by treating each individual as an independent dataset. This could be more informative when assessing the reliability of the different methods, and may highlight potential anomalies in the data for a given sample arising, for instance, from differences in museum treatment, unintended generation of artefacts during molecular work, or symptomatic of aDNA degradation. Presumably this has been performed, but not reported. In either case, it would be interesting to know if there is any difference in within-sample reproducibility of the different methods, in samples of different age.

Reviewer 2 ·

Basic reporting

Overall the manuscript seemed well presented and coherent.

Experimental design

Experimental design appeared to me to be logical and clearly explained.

Validity of the findings

The conclusions with respect to methodology seemed fine.
I do think it possibly a bit early to claim that the study confirms that "in general, koalas share few integration sites among individuals".

Reviewer 3 ·

Basic reporting

I have some doubts about the structure of the paper. I mean, it is impressive the quantity of the data and the methods are thoroughly explained. However, it is overwhelming and it makes the reading of the paper hard. In addition, some parts of materials and methods sounded results (lines 277-278, 310-314), some results discussion (lines 390-391, 444-446, 464, 479-484) and part of the discussion a recapitulation of results (lines 530-534). Indeed, I am not sure if the labels of images and tables should be so extensive and there are too much tables/figures in material and methods. Finally, I do not understand why the biologically most important finding is in the supplementary article, it is the “evidence” of the validity of the approach proposed. I think it deserves being in the main manuscript. Thus, I think that the manuscript should be reorganized to be more clear and to be adjusted to the message that it is want to send: is it merely a comparison of method? Centre the results and discussion on it and the pros and caveats of the approaches; is it the successful evidence of retroviral insertion profiling in ancient like DNA? Put the light on it. That is, match the narrative to the message.

Experimental design

The experimental design is impressive. However, I have two minor concerns:
- The polymorphism and degradation of LTRs. They discussed this problem in lines 514-518 and they think carefully an approach to avoid the biases. However, what I miss is the mapping of the reads from the modern Koala and the reads from ancient Koalas against the genomes of KoERVs deposited in Genbank (completes and/or LTRs) and try to estimate the number and positions of the variants. And based on this data they could optimize the analyses explained in lines 246-254.
- The use of BLAST with short reads. It is a crude option to deal with short reads, I would prioritize results based in mapping and, if this approach has not been used in a given analysis, try an alternative. For example, use the subject database as a synthetic genome to be mapped and the queries the reads to be mapped.

Validity of the findings

Since the comparison of insertion site between koalas is in the supplemental article (be careful with leaving the comments/track changes on) is not a main discussion and conclusion, but it is interesting as I stated before. Since there are common integration sites between modern and “ancient” koalas, the presence/absence in geography and time could be a valuable finding and it could point out the utility of the approach used in the paper. Everything else is fine.

Comments for the author

I am aware that there is a lot of work and it is difficult to summarize it. However, it is a pity that this hard work could be vanished between so much data. I encourage the authors to work on the text to be clearer.

---

## Round 0.2 · accepted · Accept

· Academic Editor

Accept

Thank you for the comprehensive responses. I am happy to accept the paper.